# Mitogenome of the stink bug *Aelia fieberi* (Hemiptera: Pentatomidae) and a comparative genomic analysis between phytophagous and predatory members of Pentatomidae

Qianquan Chen[ID]◎, Yongqin Li◎, Qin Chen, Xiaoke Tian, Yuqian Wang, Yeying Wang[ID]*

School of Life Sciences, Guizhou Normal University, Gui'an, China

◎ These authors contributed equally to this work.
* yeyingwang@gznu.edu.cn

## Abstract

*Aelia fieberi* Scott, 1874 is a pest of crops. The mitogenome of *A. fieberi* (OL631608) was decoded by next-generation sequencing. The mitogenome, with 41.89% A, 31.70% T, 15.44% C and 10.97% G, is 15,471 bp in size. The phylogenetic tree showed that Asopinae and Phyllocephalinae were monophyletic; however, Pentatominae and Podopinae were not monophyletic, suggesting that the phylogenetic relationships of Pentatomoidae are complex and need revaluation and revision. Phytophagous bugs had a ~20-nucleotide longer in *nad2* than predatory bugs. There were differences in amino acid sequence at six sites between phytophagous bugs and predatory bugs. The codon usage analysis indicated that frequently used codons used either A or T at the third position of the codon. The analysis of amino acid usage showed that leucine, isoleucine, serine, methionine, and phenylalanine were the most abundant in 53 species of Pentatomoidae. Thirteen protein-coding genes were evolving under purifying selection, *cox1*, and *atp8* had the strongest and weakest purifying selection stress, respectively. Phytophagous bugs and predatory bugs had different evolutionary rates for eight genes. The mitogenomic information of *A. fieberi* could fill the knowledge gap for this important crop pest. The differences between phytophagous bugs and predatory bugs deepen our understanding of the effect of feeding habit on mitogenome.

## Introduction

Stink bugs (Hemiptera: Pentatomidae) are known to excrete odorous bodily fluids as a defensive strategy when disturbed [1]. They are also known as shield bugs because of their shield shape. Currently, 4722 species of stink bugs have been identified around the world, which were divided into ten subfamilies and 896 genera (Pentatomoidea home page, https://www.ndsu.edu/faculty/rider/Pentatomoidea/Classification/Genus_Species_Numbers.htm). The ten subfamilies are Aphylinae (three species), Asopinae (299 species), Cyrtocorinae (11 species), Discocephalinae (303 species), Edessinae (306 species), Pentatominae (3336 species),

**Data Availability Statement:** The genome sequence data that support the findings of this study are openly available in GenBank of NCBI at

(https://www.ncbi.nlm.nih.gov/) under the accession no. OL631608.

**Funding:** This research was funded by National Natural Science Foundation of China, grant number 32060124; Guizhou Normal University, grant number Qianshixinmiao [2021]A11. The funders had no role in study design, data collection and analysis, decision to publish, or preparation of the manuscript.

**Competing interests:** The authors have declared that no competing interests exist.

Phyllocephalinae (213 species), Podopinae (249 species), Serbaninae (one species) and Stirotarsinae (one species). Based on morphological characteristics and DNA sequences, including 18S rRNA, 16S rRNA, 28S rRNA and *cox1*, Pentatomidae were considered monophyly [2]. Additionally, the morphological characteristics supported these ten subfamilies were monophyletic [3]. However, morphological characteristics exhibit variations depending on habits, creating barriers to classification [4]. A molecular phylogeny tree constructed with a combination of mitochondrial (*cox1* and 16S RNA) and nuclear genes (18S RNA and 28S RNA) showed that Pentatomidae were not monophyletic [5]. Furthermore, a molecular phylogenetic tree constructed with mitogenomic protein-coding genes (PCGs) showed that both Pentatominae and Podopinae were not monophyletic [6], suggesting that the taxonomic classification of Pentatomidae needed revaluation and revision. In short, the relationships of the stink bugs have remained far from solved.

Most of the species of Pentatomidae are phytophagous. They cause tremendous economic damage to crops throughout the world [7, 8]. For example, *Halyomorpha halys* could reduce soybean yield by 50% in the mid-Atlantic region [9]. Owing to its ability to adapt to a new environment, it is considered an important invasive stink bug. In addition to the damage posed by invasive stink bugs, the damage caused by native stink bugs cannot be ignored. For example, *Aelia fieberi* Scott, 1874 feeds on rice, wheat, and grasses [10]. Under long-day photoperiod, the female develops fast to lay eggs. It has two generations per year and its adults enter diapause to survive winter [10]. Very few species of Pentatomidae are non-phytophagous, i.e., predatory stink bugs, which prefer to feed on large and soft bodied prey. Most species of Asopinae (Heteroptera: Pentatomidae) prey on the larvae of some species of Lepidoptera, Coleoptera, and Hemiptera [6]. Therefore, some species of Asopinae are used as biological control agents against agricultural pests [11]. Compared to phytophagous stink bugs, predatory stink bugs have a more robust and stouter labium [12].

Mitochondria provide energy to the eukaryotic cell through oxidative phosphorylation. The proteins involved in oxidative phosphorylation are encoded by the nuclear and mitochondrial genomes. The animal mitochondrial genome (mitogenome) is a circular double-stranded DNA molecule. Generally, it encodes 13 PCGs, two ribosomal RNAs (rRNAs), and 22 transfer RNAs (tRNAs) [13]. In addition, it contains a noncoding control region (AT rich region), which contains essential regulatory elements for replication and transcription [14]. Owing to the lack of protection of histones, mitogenomic DNA has a mutation frequency much higher than nuclear DNA. As a result, mitogenomes are widely used in phylogenetics, population genetics, and evolutionary biology [15, 16]. With the development of next-generation sequencing (high-throughput sequencing), many mitogenomes have been decoded using this technique. Phytophagous and predatory stink bugs have differences in nutrients, which can result in differences in metabolism between them. The effect of feeding habits on the mitogenome remains unknown.

In this work, the mitogenome of *A. fieberi* has been decoded. The characteristics of the mitogenome, including nucleotide composition, codon usage, secondary structure of tRNA, and the evolutionary rate of PCGs, were systematically analyzed. The phylogenetic tree of Pentatomidae was constructed with the new mitogenome and 52 mitogenomes extracted from NCBI. Furthermore, the effect of feeding habits on the mitogenome was explored.

## Materials and methods

### Samples and identification

*Aelia fieberi* is a common pest of developing crops in China and is not recorded in the species list of the ethics committees for research involving animals of the Guizhou Normal University.

Therefore, no ethical approval or other relevant permission can be provided for the study. The *A. fieberi* samples were collected from the campus of Guizhou Normal University (26˚ 22′50.30″N, 106˚38′11.72″E) in April 2019. All samples were identified from their morphological characteristics. The total DNA was then extracted from the muscle tissue of an adult sample using the phenol-chloroform extraction [17, 18]. The mitochondrial cytochrome c oxidase subunit I gene (*cox1*) was amplified with primers (LCO1490 and HCO2198) [19]. The PCR productions were checked by agarose gel electrophoresis and sequenced by Sanger sequencing. The sequence was submitted to BOLD systems v4 (http://www.boldsystems.org/) as a query for species identification. The similarity between the query sequence and the reference sequence of *A. fieberi* in the BOLD was 99.65%. The specimens (specimen ID: GZNU-cqq-8) were soaked in absolute alcohol and then stored at 4˚C in the Museum of Guizhou Normal University.

## Next-generation sequencing, annotation, and bioinformatics analysis

Total DNA was isolated from muscle tissue from an adult specimen with the ONE-4-ALL Genomic DNA Mini-Prep Kit (BS88504, Sangon, Shanghai, China). DNA was fragmented, then ~500 bp DNA was recycled. Paired-end libraries were constructed with the Illumina platform. DNA was sequenced using Illumina Hiseq X ten at Sangon Biotechnology Company (Shanghai, China). The adapter sequences were removed and low-quality reads were trimmed with Trimmomatic (v0.36) [20]. The mitogenome of *Dolycoris baccarum* (NC_020373) was used as a reference and clean reads were assembled with SOAPdenovo2 (v2.04) [21]. PCGs were identified by BLAST comparison with the *D. baccarum* mitogenome. Secondary structures of tRNAs were predicted with MITOS2 and tRNAscan-SE [22, 23], and rRNAs and noncoding control region (AT-rich region) were determined by the boundary of tRNAs. The nucleotide composition was calculated with MEGA X [24]. AT skew values were calculated using the following formula: AT skew = (A—T)/(A + T). Similarly, GC skew values were calculated by GC skew = (G—C)/(G + C). Codon usage indices, including codon count, the relative synonymous codon usage (RSCU), were calculated with MEGA X [24]. The number of nonsynonymous substitutions per nonsynonymous site (Ka), and the number of synonymous substitutions per synonymous site (Ks) were calculated with DnaSP 6 [25]. A circular map of the mitogenome of *A. fieberi* was generated with the CGView server [26]. The amino acid sequence logos were created with WEBLOGO (http://weblogo.berkeley.edu/) [27]. The heatmaps of codon usage and amino acid usage were generated with ggplot2 as implemented in R (v4.1.2). Figures were edited with Adobe Illustrator CS5.

## Phylogenetic analysis

The mitogenomes of *A. fieberi* and 52 complete mitogenomes available from Pentatomidae in NCBI were used to construct the phylogenetic tree of Pentatomidae. *Eurygaster testudinaria* (NC_042808, Hemiptera: Scutelleridae) was selected as representative of the outgroup. These mitogenomes were imported into PhyloSuite (v1.2.3) [28]. Then the nucleic acid sequences of 13 PCGs were extracted from these mitogenomes. Codon-based multiple alignments were performed with MAFFT as implemented in PhyloSuite [28]. Then the alignments of 13 PCGs were concatenated. PartitionFinder2 was used to select the best-fit partitioning strategy and models for the concatenated sequences. Phylogenetic trees were reconstructed by the Bayesian method (Mrbayes v3.2.6) and maximum likelihood (IQ-TREE v1.6.8) as implemented in PhyloSuite [28]. The number of generations was 200, 000 and partition models were selected as models [28]. The phylogenetic tree was visualized with Figtree (v1.4.4) and Adobe Illustrator CS5.

## Results

### Genomic structure and nucleotide composition

The complete mitogenome of *A. fieberi* (OL631608) was 15,471 bp, which encoded 13 PCGs, 22 tRNAs, two rRNAs, and a noncoding control region (Fig 1, S1 Table). Fourteen genes, including four PCGs, eight tRNAs, and two rRNAs, were encoded by the minority strand (L strand), and the remaining genes were encoded by the majority strand (H strand). The arrangement of the genes of *A. fieberi* was in accordance with that of other species of Pentatomidae such as *Eysarcoris guttigerus* [18]. Pentatomidae species with a complete mitogenome sequence have the same gene arrangement in the mitogenome, except *Priassus spiniger* (OK546352). The arrangement of genes rotated around the 19th arginine tRNA in *P. spiniger* (S1 Fig). A total of 130-bp intergenic spacers were distributed across 20 locations. The shortest intergenic spacers were one bp, and the longest intergenic spacer, located between *trnS2* and *nad1*, was 24 bp. A total of 25 bp overlaps were distributed across five locations. The shortest

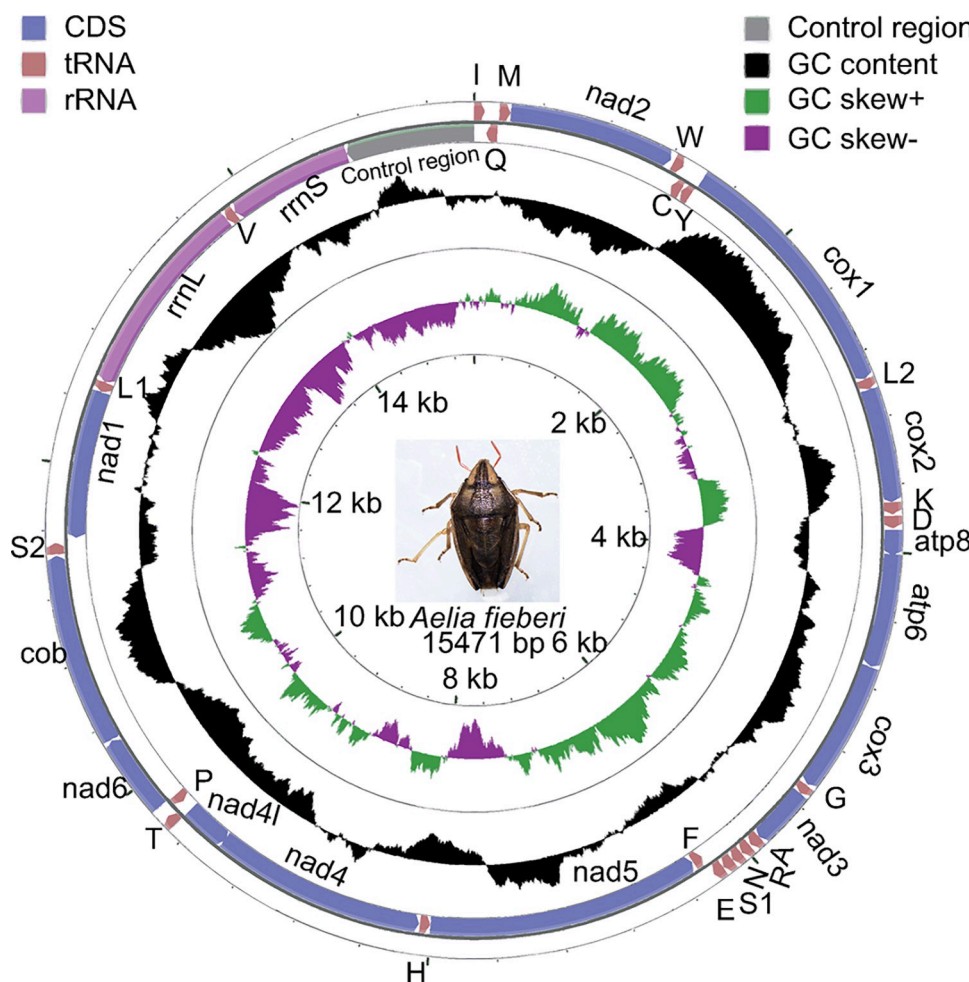

**Fig 1. Mitogenome map of *Aelia fieberi*.** Protein-coding genes (PCGs, CDS) and ribosomal genes (rRNAs) are presented with standard abbreviations. Genes coding for transfer RNAs (tRNAs) are presented with one letter abbreviation. S1 = AGN, S2 = UCN, L1 = CUN, L2 = UUR. Wathet blue, orange, pink and grey represent PCGs, tRNAs, rRNAs and D-loop (noncoding control region), respectively. The colors black, green, and purple represent the GC content, positive GC skew (GC skew+) and negative GC skew (GC skew-), respectively. The orientation of the gene is indicated by arrows.

overlap was one bp, which was located between *trnN* and *trnS1*. The longest overlap, located between *trnW* and *trnC*, was eight bp. The noncoding control region, located between *rrnS* and *trnI*, was 803 bp.

The mitogenome consisted of 41.89% A, 31.70% T, 15.44% C, and 10.97% G (S2 Table). The AT content of the whole genome, PCGs, tRNAs, rRNAs and noncoding control region was 73.59%, 73.93%, 74.79%, 76.71% and 71.73%, respectively. The AT skew value was 0.14, and the GC skew value was -0.17, indicating that the mitogenome had a preference for A and C.

## Protein-coding genes

Among 13 PCGs, *atp8* was the shortest gene (150 bp) and *nad5* was the longest gene (1702 bp). Four PCGs, including *nad1*, *nad4*, *nad4l*, and *nad5*, were encoded by the L strand, and the remaining nine PCGs were encoded by the H strand (Fig 1, S1 Table). In *A. fieberi*, *nad1* used TTG as the initiation codon; however, the remaining 12 PCGs used ATN as the initiation codon (ATA for *cox1*, *cox2*, *atp8*, *nad3*, and *nad6*; ATG for *atp6*, *cox3*, *nad4*, and *cob*; ATT for *nad5* and *nad4l*; ATC for *nad2*). *Nad1* used TAG as a stop codon; however, *cox1*, *cox2*, *nad3*, and *nad5* used T as an incomplete stop codon (S1 Table). The remaining eight PCGs used TAA as a stop codon.

Relative synonymous codon usage (RSCU) analysis of *A. fieberi* showed that a total of 61 codons were used, except for two stop codons (TAA and TAG) (S3 Table). A codon (AGG (S)) was not used in *A. fieberi*. Serine had eight codons, leucine had six codons, and six amino acids, including valine, proline, threonine, alanine, arginine, and glycine, had four codons, and the remaining amino acids had two codons (S3 Table). The highly frequent codons were UUU (F), AUA (M), UUA (L) and AUU (I), which accounted for 30.80% of the total codons used by the mitogenome. The RSCU values of five codons, including CGA (R), UCU (S), UCA (S), AGA (S) and UUA (L), were more than two (S3 Table). On the contrary, codons with high GC content, including GCG (1), CGG (6), and AGC (6) were the least frequent codons.

## Transfer RNAs

The size of the tRNAs ranged from 63 to 72 bp (S1 Table). Among the 22 tRNAs, *trnG* and *trnR* were the shortest tRNAs and *trnD* was the longest tRNA. Eight tRNAs, including *trnH*, *trnY*, *trnL1*, *trnC*, *trnP*, *trnF*, *trnV*, and *trnQ*, located on the L strand, the remaining tRNAs located on the H strand (Fig 1, S1 Table). The distribution pattern of the tRNAs in the mitogenome was in accordance with that of other species of Pentatomidae. The AT content ranged from 65.15% (*trnM*) to 84.06% (*trnE*) (S2 Table). The AT skew ranged from -0.27 (*trnY*) to 0.16 (*trnM*), and the GC skew ranged from -0.27 (*trnW*) to 0.56 (*trnH*). *TrnS1* lacked a dihydrouridine (DHU) arm (Fig 2). The remaining tRNAs had a typical cloverleaf secondary structure. The non-Watson-Crick base pairing (G-U pairing) appeared in the acceptor stem of *trnY*, *trnK*, *trnA*, *trnF*, and *trnH*. G-U pairing also appeared in the anticodon arm of *trnE* and *trnH*, the DHU arm of *trnY*, *trnG*, *trnH*, and *trnP*, and the TψC arm of *trnQ*, *trnA*, and *trnS1*. Three tRNAs, including *trnC*, *trnT*, and *trnS2*, used U as the discriminator nucleotide, while *trnH* used C as the discriminator nucleotide. The remaining tRNAs used A as the discriminator nucleotide (Fig 2).

## Phylogenetic relationships

Phylogenetic analyses were performed with 13 PCG nucleotide sequences extracted from 53 mitogenomes of Pentatomidae. The Bayesian tree showed that *Neojurtina typica* had a sister group relationship with the remaining 52 species (Fig 3). *Aelia* had a closer relationship with

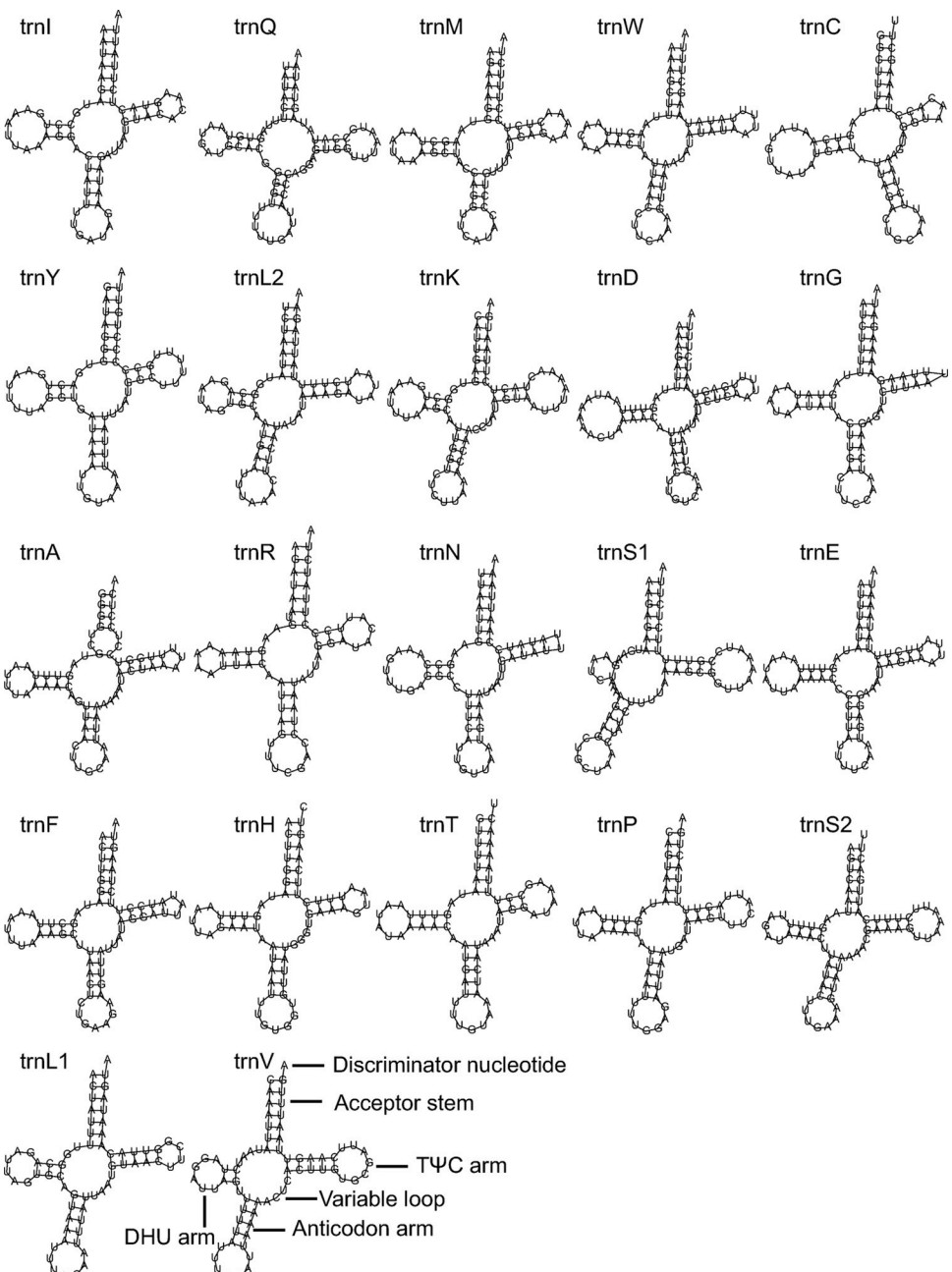

**Fig 2. Predicted secondary structure of 22 tRNA of *A. fieberi*.**

*Dolycoris* than the remaining genera. *Aelia*, *Dolycoris*, *Anaxilaus*, *Glaucias*, *Plautia*, *Nezara* and *Palomena* were clustered into a clade. Their relationship was ((*Aelia* + *Dolycoris*) + ((*Nezara* + *Palomena*) + ((*Plautia* + *Glaucias*) + *Anaxilaus*))). Predatory bugs (Asopinae, in red) were clustered into a clade. Their relationship was ((((*Picromerus* + *Eocanthecona*) + *Dinorhynchus*) + *Cazira*) + (*Arma* + *Zicrona*)). Asopinae and Phyllocephalinae were monophyletic, however, Pentatominae and Podopinae were not monophyletic. For Podopinae species, *Dybowskyia reticulata* was clustered with *Graphosoma rubrolineatum*; however, *Scotinophara lurida* and *Deroploa parva* were clustered with *Catacanthus incarnatus*

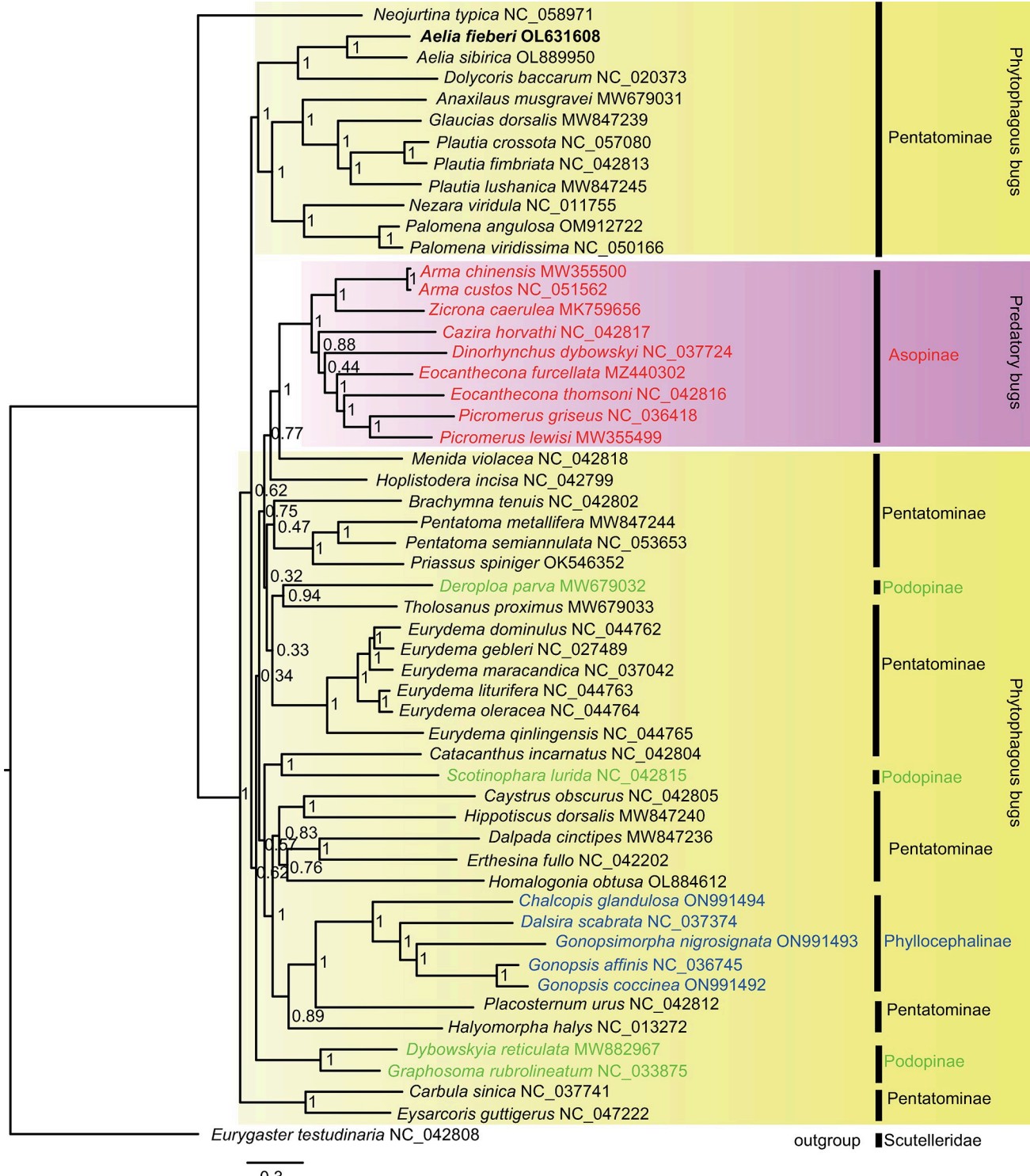

**Fig 3. Bayesian phylogenetic tree of 53 species of Pentatomidae.** *Eurygaster testudinaria* (Hemiptera: Scutelleridae) was selected as representative of the outgroup. The posterior probabilities were labeled at each node. GenBank accession numbers of sequences were listed after the species name.

(Pentatominae) and *Tholosanus Proximus* (Pentatominae), respectively. The maximum likelihood tree had a similar topology to the Bayesian tree (S2 Fig).

## The effect of feeding habit on mitogenome

Compared to predatory bugs, phytophagous bugs had a ~20-nucleotide longer in *nad2* (Fig 4). There were differences in amino acid sequence at six sites between predatory bugs and phytophagous bugs (Fig 5). Specifically, for *nad2*, phytophagous bugs used isoleucine at the 165[th] site, predatory bugs used leucine; for *nad3*, phytophagous bugs used leucine at the 80[th] site, predatory bugs used isoleucine; for *nad4*, phytophagous bugs used phenylalanine at the 440[th] site, predatory bugs used isoleucine; for *nad4L*, phytophagous bugs used methionine at the 12[th] site, predatory bugs used leucine; for *nad5*, phytophagous bugs used tyrosine at the 383[th] site, predatory bugs used phenylalanine; for *cox1*, phytophagous bugs used threonine at the 489[th] site, predatory bugs used methionine.

For the initiation codon, ATG, ATT, ATA, TTG, AAA, AAC, ATC, and GTG accounted for 33.33%, 20.85%, 20.00%, 22.39%, 0.17%, 0.17%, 2.74%, and 0.34% in phytophagous bugs, respectively (S3A Fig). ATG, ATT, ATA, TTG, ATC and TTA accounted for 34.19%, 23.93%, 21.37%, 18.80%, 0.85% and 0.85% in predatory bugs (S3B Fig). For the stop codon, TAA, T, TAG and TA accounted for 68.38%, 24.62%, 6.50%, and 0.51% in phytophagous bugs (S3C Fig). TAA, T, and TAG accounted for 74.36%, 18.80%, and 6.84% in predatory bugs (S3D Fig). Together, there was no obvious difference in the use of initiation codons and stop codons between phytophagous bugs and predatory bugs. Generally, codons with A or U at the third position were used more frequently than those with G or C at the third position for both phytophagous bugs and predatory bugs (Fig 6A). The effective number of codons had a liner relationship with the total GC content of the codons (S4A Fig) and GC content of the third position of the codons (S4B Fig) in *A. fieberi*. A similar relationship occurred in other species of Pentatomidae with a complete mitogenome sequence (S4C and S4D Fig). The most abundant amino acids were leucine, isoleucine, serine, methionine, and phenylalanine (Fig 6B). Amino acids including arginine, cysteine, and glutamine were the least frequently used. On the whole, there were no obvious differences in codon usage and amino acid usage between phytophagous bugs and predatory bugs.

Compared to the Ks value, the Ka value had obvious variations among 13 PCGs (Fig 7). *Cox1* and *atp8* had the lowest and highest Ka values, respectively (Fig 7). Consequently, *Cox1* and *atp8* had the lowest and highest Ka/Ks values, respectively. The Ka/Ks value for 13 PCGs was less than 1, which indicated that all PCGs were evolving under purifying selection. Compared to phytophagous bugs, predatory bugs had higher Ks values for *atp8*, *nad2*, and *nad5*. Predatory bugs had higher Ka values for *atp6*, *nad1*, *nad2*, *nad3*, *nad4l*, and *nad5* than phytophagous bugs. As a result, predatory bugs had higher Ka/Ks values for *atp6*, *atp8*, *nad1*, *nad2*, *nad3*, *nad4*, *nad4l*, and *nad5* than phytophagous bugs.

## Discussion

In a previous study, the partial mitogenome of *A. fieberi* (12024 bp) (MK251125.1) was decoded [29], however, the sequence of *trnI*, *trnQ*, *trnM*, *nad2*, *rrnL* (partial), *trnV*, *rrnS* and noncoding control region was missing. Furthermore, a 107-nucleotide sequence in *nad5* was not decoded. As a result, the partial mitogenomic sequence was not annotated in NCBI. In the study, the complete mitogenome of *A. fieberi* (OL631608) was successfully decoded. Currently, for Pentatomidae species, the mitogenome of *Eocanthecona thomsoni* (NC_042816) is the smallest (14782 bp) [30], and the mitogenome of *Picromerus lewisi* (NC_058610) is the largest (19587 bp). The size of the mitogenome of *A. fieberi* (15471 bp) is relatively small in species of

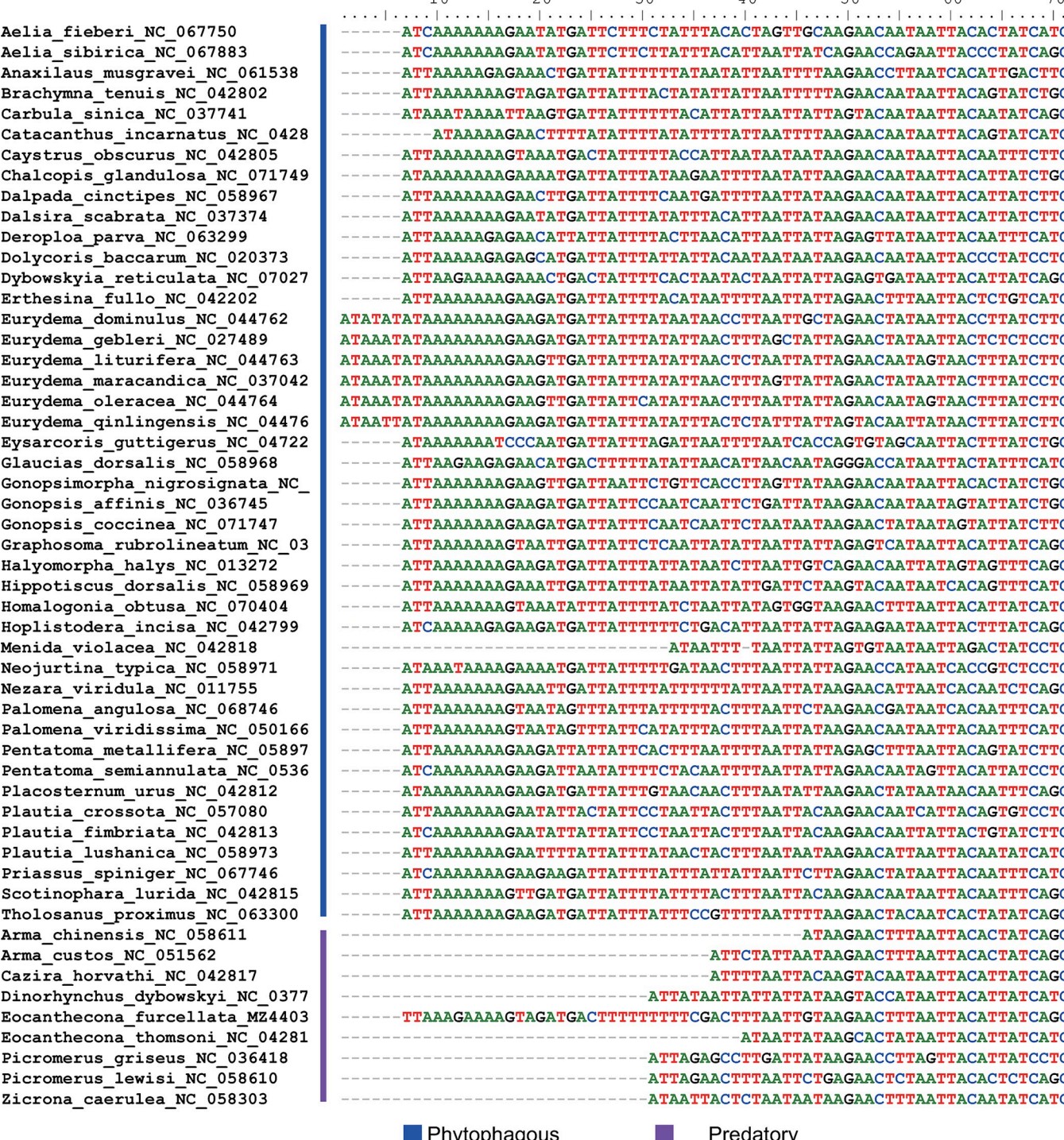

**Fig 4. Comparison of the nucleotide sequence of *nad2* between phytophagous bugs and predatory bugs.**

Pentatomidae. The order of these genes in the mitogenome of *A. fieberi* is consistent with that of most bugs such as *E. guttigerus* [18]. The sequence of *P. spiniger* (OK546352) has been submitted to GenBank, however, the sequence has not been checked by Sanger sequencing, and corresponding results have not been published. As a result, the gene arrangement in *P. spiniger*

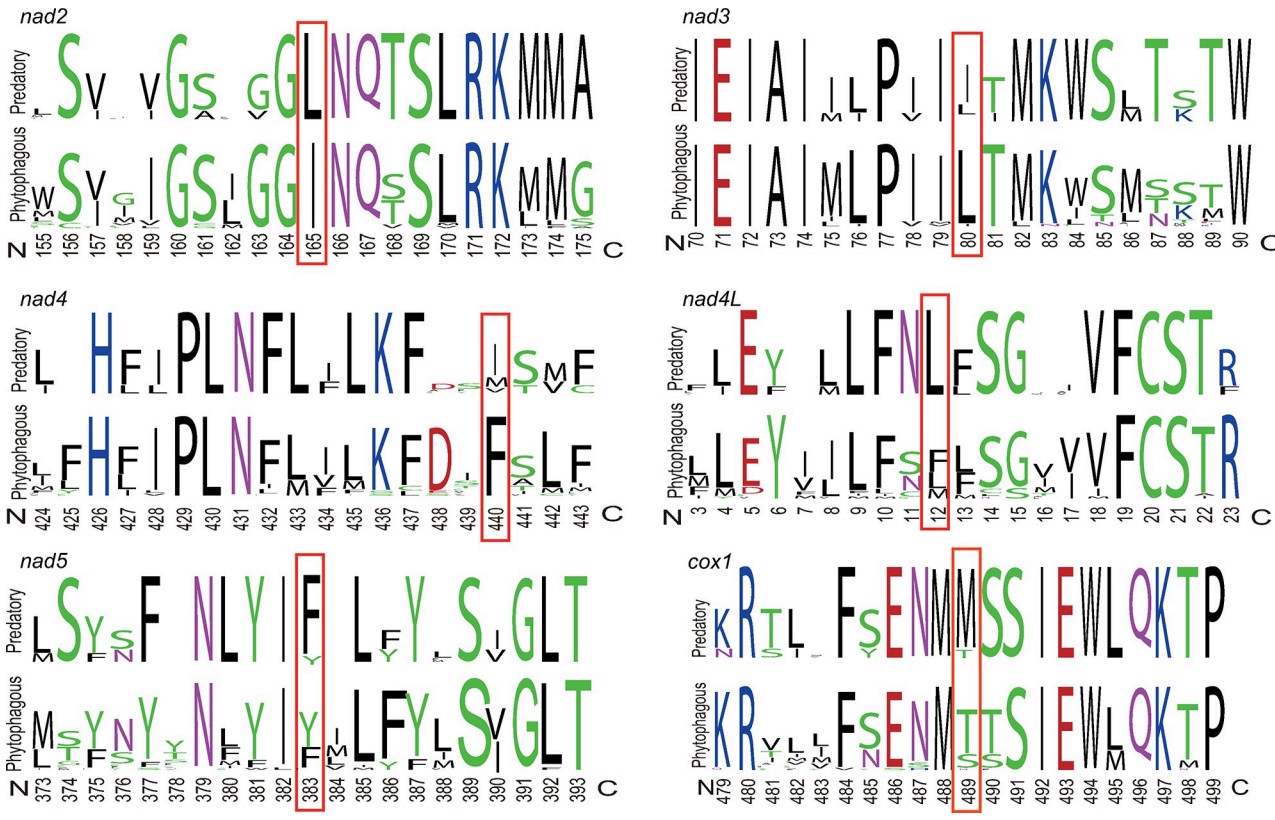

**Fig 5. Comparison of amino acid sequence at six sites between phytophagous bugs and predatory bugs.**

needs further Sanger sequencing to prove its correctness. For the species of Pentatomidae with complete mitogenomic information, *Picromerus griseus* (NC_036418) has the lowest AT content (71.70%), and *Gonopsis coccinea* (NC_071747) has the highest AT content (78.90%). The AT content of *A. fieberi* (73.60%) seems to be relatively low in these species.

The position of the noncoding control region in *A. fieberi* was in accordance with that of other species of Pentatomidae [18]. The AT content of the noncoding control region accounted for 71.73%, which was located in the range of PCG (67.08% - 82.00%) and tRNA (65.15% - 84.06%), suggesting that the name of AT rich region did not suit the noncoding control region. Currently, for species of Pentatomoidae with complete mitogenome sequences, the length of the control region ranges from seven bp (*Eocanthecona thomsoni* (NC_042816)) to 2038 bp (*Eurydema dominulus* (NC_044762)) [31, 32]. Longer than *E. thomsoni*, *Hoplistodera incisa* (NC_042799) (185 bp) and *Carbula sinica* (NC_037741) (643 bp) rank second and third, respectively. More and more mitogenomes are decoded by high-throughput sequencing, and partial noncoding control region is usually missing. Generally, there are tandem repeats in the noncoding control region [31], however, no tandem repeat was detected in the noncoding control region of *A. fieberi*. Similarly, no tandem repeats were detected in some Hemiptera species such as *Notobitus montanus*, *Hydaropsis longirostris*, and *Pseudomictis tenebrosa* [32, 33].

In *A. fieberi*, only *nad1* used TTG as the initiation codon, and the remaining 12 PCGs used ATN as the initiation codon. TTG is usually used as an initiation codon for *cox1* in animals [34–37]. However, TTG is used as an initiation codon for *atp8*, *nad1*, *nad4l*, *nad5* and *nad6* in several insects [18, 31, 32]. Nine PCGs used TAN as a stop codon, and the remaining four

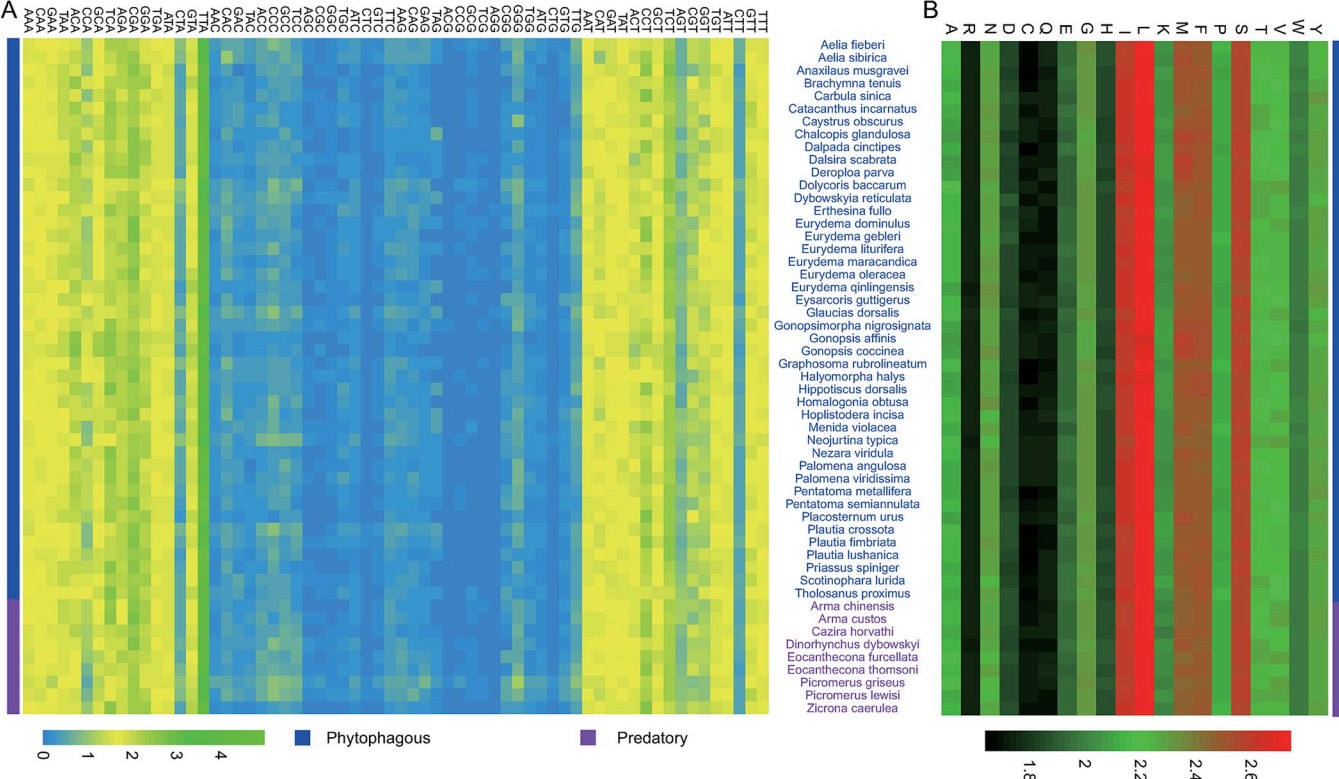

**Fig 6. Use of coding and amino acids.** (A) Heatmap indicates codon usage in 53 species of Pentatomidae. (B) Heatmap indicates amino acid usage in 53 species of Pentatomidae.

PCGs, including *cox1*, *cox2*, *nad3*, and *nad5*, used T as an incomplete stop codon. Incomplete stop codons (TA or T) are frequently used in metazoan mitogenomes, which could be added with A or AA by post-transcriptional polyadenylation [38–40]. RSCU analysis indicated that each codon was not used at the same frequency. For 53 species of Pentatomidae, they prefer to use codons that used A or U at the third position of the codon than those that used G or C at the third position of the codon, suggesting that the GC content of codons could influence codon usage [33]. The analysis of amino acid usage indicated that leucine, isoleucine, serine, methionine, and phenylalanine were the most abundant. Similarly, in Coreidae (Hemiptera) species, leucine, serine, phenylalanine, isoleucine, and tyrosine were the most abundant [33], suggesting that leucine, isoleucine, serine, and phenylalanine were relatively abundant in Hemiptera, the abundance of the remaining amino acids had variation between different families.

The variation range AT content of tRNA (65.15% - 84.06%) were slightly higher than those of PCG (67.08% - 82.00%). For tRNAs, the AT skew ranged from -0.27 (*trnY*) to 0.16 (*trnM*), and the GC skew ranged from -0.27 (*trnW*) to 0.56 (*trnH*). For PCGs, the AT skew ranged from -0.34 to 0.14, and the GC skew ranged from -0.22 to 0.32. It seemed that the variation range of the AT skew of tRNAs was a little smaller than that of PCGs; however, the variation range of the GC skew of tRNAs was much larger than that of PCGs. Most tRNAs had the typical cloverleaf secondary structure, however, *trnS1* lacked a dihydrouridine (DHU) arm. The phenomenon of *trnS1* lacking a DHU arm is popular in insects [33–34, 40]. Three tRNAs (*trnC*, *trnT* and *trnS2*) and *trnH* used U and C as discriminator nucleotides, respectively. The remaining 18 tRNAs used A as the discriminator nucleotide. Similarly, *trnC*, *trnT*, *trnS2* and

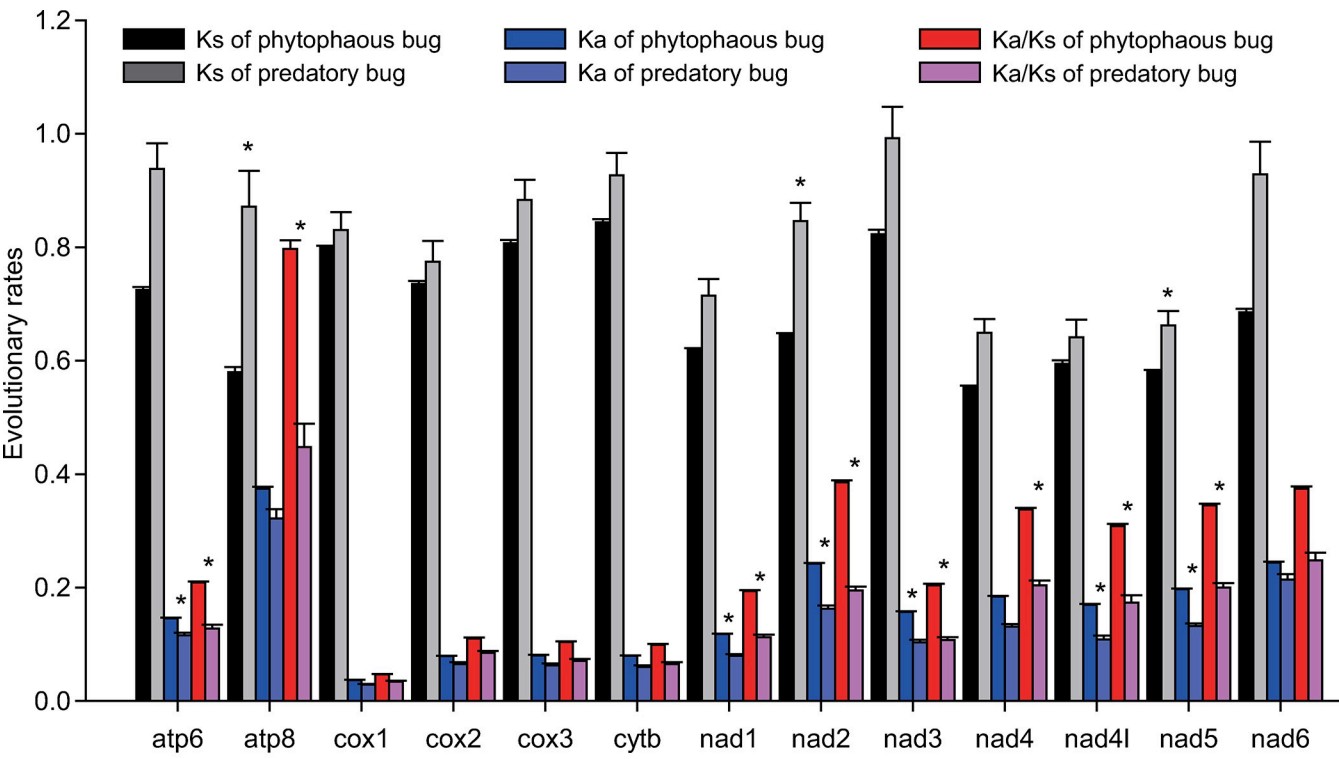

**Fig 7. Comparison of the evolutionary rates of 13 protein-coding genes between phytophagous bugs and predatory bugs.** Ks represents synonymous nucleotide substitutions per synonymous site; Ka represents nonsynonymous nucleotide substitutions per nonsynonymous site; Ka/Ks represents the ratio of Ka to Ks. The error bars presented the standard error of the mean.

*trnH* used U as a discriminator nucleotide in Hemiptera species [33, 40]. Non-Watson-Crick base pairing (G-U pairing) appeared in the acceptor stem, anticodon arm, DHU arm, and TψC arm of tRNAs. The G-U pairing could increase the stability of tRNAs, which is popular in insects [15].

Currently, 4722 species of Pentatomidae have been divided into ten subfamilies. However, only 53 complete verified mitogenomes of Pentatomidae species are available in NCBI. Four, five, nine, and 35 complete mitogenomes were decoded for Podopinae (249 species), Phyllocephalinae (213 species), Asopinae (299 species), and Pentatominae (3336 species), respectively. Currently, there is no complete mitogenomic information for the remaining six subfamilies, including Aphylinae (three species), Cyrtocorinae (11 species), Discocephalinae (303 species), Edessinae (306 species), Serbaninae (one species), and Stirotarsinae (one species). The Ka/Ks values for 13 PCGs were lower than one, suggesting that all PCGs could be used to reconstruct the phylogenetic tree. The phylogenetic tree showed that 53 species were divided into four subfamilies. Asopinae and Phyllocephalinae were monophyletic; however, Pentatominae and Podopinae were not monophyletic. Based on mitochondrial (*cox1* and 16S RNA) and nuclear genes (18S RNA and 28S RNA), Pentatominae and Podopinae were not monophyletic [5]. Furthermore, evidence from mitogenomes supported that both Pentatominae and Podopinae

were not monophyletic [6]. On the contrary, the morphological characteristics supporting these ten subfamilies were all monophyletic [3]. Owing to the morphological characteristics that exhibit variations depending on habits, the results of molecular phylogenetic analysis might have more credibility [4]. The Bayesian tree showed that *Aelia* had a closer relationship with *Dolycoris* than the remaining genera. The predatory bugs (Asopinae) clustered in a clade. For Podopinae species, *Dybowskyia reticulata* was clustered with *Graphosoma rubrolineatum*, however, *Scotinophara lurida* and *Deroploa parva* were clustered with *Catacanthus incarnatus* (Pentatominae) and *Tholosanus Proximus* (Pentatominae), respectively. The Pentatominae had a sister group relationship with Asopinae, Podopinae, and Phyllocephalinae, suggesting that the phylogenetic relationship of the Pentatominae was complex. In short, the phylogenetic relationships of stink bugs have remained far from solved. The mitogenomic information of *A. fieberi* could fill the knowledge gap for this important crop pest.

Phytophagous bugs had a ~20-nucleotide longer in *nad2* than predatory bugs (Fig 4). The ~20-nucleotide could be used to design small interfering RNA (siRNA) to manage phytophagous bugs. There were differences in amino acid sequence at six sites between predatory bugs and phytophagous bugs (Fig 5). In addition to *nad3*, the remaining five genes, including *nad2*, *nad4*, *nad4l*, *nad5*, and *cox1*, preferred to use highly abundant amino acids in predatory bugs, which could improve the translation efficiency. Furthermore, the properties of amino acid might influence the activity of enzymes, which need further experiment to verify.

Among the 13 PCGs, *cox1* had the lowest Ka/Ks value, and *atp8* had the highest Ka/Ks value. At the same time, *cox1* had the lowest AT content (67.08%) and *atp8* had the highest AT content (82.00%), suggesting that the evolutionary rates of genes were influenced by the AT content. Predatory bugs had a higher Ka/Ks value of *atp6*, *atp8*, *nad1*, *nad2*, *nad3*, *nad4*, *nad4l* and *nad5* than phytophagous bugs, suggesting that these genes had stronger purifying selection stress in phytophagous bugs than predatory bugs.

## Conclusions

The study has decoded the complete mitogenome of *A. fieberi*. The phylogenetic tree showed that Asopinae and Phyllocephalinae were monophyletic and Pentatominae and Podopinae were not monophyletic. The codon usage analysis indicated that highly used codons used either A or U at the third position of the codon. The analysis of amino acid usage showed that leucine, isoleucine, serine, methionine, and phenylalanine were the most abundant in 53 species of Pentatomoidae. Phytophagous bugs had a ~20-nucleotide longer in *nad2* than predatory bugs. Phytophagous bugs and predatory bugs used different amino acid at six sites. All PCGs were evolving under purifying selection, *Cox1* and *atp8* had the strongest and lowest purifying selection stress, respectively. Phytophagous bugs and predatory bugs had a different evolutionary rate for eight genes.

## Supporting information

**S1 Fig. Gene arrangement in *Priassus spiniger*.**
(TIF)

**S2 Fig. Maximum likelihood phylogenetic tree of 53 species of Pentatomidae.** *Eurygaster testudinaria* (Hemiptera: Scutelleridae) was selected as representative of the outgroup. The bootstrap values were labeled at each node. GenBank accession numbers of sequences were listed after the species name.
(TIF)

**S3 Fig. Comparison of the start and stop codons between phytophagous bugs and predatory bugs.** (A) Use of initiation codons in phytophagous bugs. (B) Use of initiation codons in predatory bugs. (C) Stop codon usage in phytophagous bugs. (D) Stop codon usage in predatory bugs.
(TIF)

**S4 Fig. The effective numbers of codons have a positive correlation with the G+C content of codons.** (A) Total G+C content of codons in *Aelia fieberi*. (B) G+C content of the third position of the codon in *A. fieberi*. (C) Total G+C content of codons in 52 species of Pentatomidae. (D) G+C content of the third position of the codon in 52 species of Pentatomidae.
(TIF)

**S1 Table. Annotation of the *Aelia fieberi* mitogenome.**
(DOCX)

**S2 Table. Nucleotide composition of *Aelia fieberi* (%).**
(DOCX)

**S3 Table. Codon usage in the mitogenome of *Aelia fieberi*.**
(DOCX)

## Author Contributions

**Conceptualization:** Qianquan Chen.

**Data curation:** Qianquan Chen, Yongqin Li, Qin Chen, Xiaoke Tian.

**Formal analysis:** Qianquan Chen.

**Funding acquisition:** Qianquan Chen.

**Investigation:** Qianquan Chen, Yongqin Li, Qin Chen, Xiaoke Tian, Yuqian Wang, Yeying Wang.

**Methodology:** Qianquan Chen.

**Project administration:** Qianquan Chen, Yeying Wang.

**Resources:** Qianquan Chen.

**Software:** Qianquan Chen.

**Supervision:** Qianquan Chen, Yuqian Wang, Yeying Wang.

**Validation:** Qianquan Chen.

**Visualization:** Qianquan Chen.

**Writing – original draft:** Qianquan Chen.

**Writing – review & editing:** Qianquan Chen.

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
