## [Decision Letter · Decision Letter 0]

24 Jul 2023

PONE-D-23-11789Mitogenome of the stink bug Aelia fieberi (Hemiptera: Pentatomidae) and a phylogenetic analysis of PentatomidaePLOS ONE

Dear Dr. Wang,

Thank you for submitting your manuscript to PLOS ONE. After careful consideration, we feel that it has merit but does not fully meet PLOS ONE’s publication criteria as it currently stands. Therefore, we invite you to submit a revised version of the manuscript that addresses the points raised during the review process. Comments on your manuscript by two reviewers and the academic editor are provided at the bottom of this email. While one reviewer has suggested rejection of the manuscript, the second reviewer and the academic editor have suggested major revision. Revise your manuscript carefully by considering the suggestions from the reviewer and the academic editor and submit the revised manuscript along with point wise replies to the comments.  

We look forward to receiving your revised manuscript.

Kind regards,

Neelesh Dahanukar, Ph.D.

Academic Editor

PLOS ONE

Journal Requirements:

"the authors declare no conflicts of interest."

Additional Editor Comments:

The study describes the whole mitogenome of the stink bug Aelia fieberi and provides comparative genomics and phylogeny of members of Pentatomidae. In general, the authors have used appropriate methods for mitogenome sequencing, assembly, annotation and comparative mitogenomic study along with phylogenetic analysis. However, there are some issues with the manuscript that authors will have to address before the manuscript can be considered further for review. Both major and minor issues are explained below in no particular order.

(1) English grammar: One of the major problems with the manuscript is the English grammar, which needs thorough editing. Authors should take help of native English speaker or professional English editing services to overcome this issue. It is impossible to provide detailed comments on English grammar as almost every other line has a problem. But just to give an example, authors mention “The phylogenetic tree showed that Asopinae and Phyllocephalinae were monophyly, however, Pentatominae and Podopinae were not monophyly.” In both the cases in the above sentence authors should have used “monophyletic” rather than “monophyly”.

(2) Mitogenome GenBank accession numbers: NCBI GenBank numbers for mitogenomes that start with "NC_" are "Provisional REFSEQ" numbers and should be replaced by the corresponding finally GenBank accession number after NCBI review. For example, the provisional refseq number for the sequence published in this manuscript is NC_067750 and the final GanBank number is OL631608. Authors should use the final GenBank numbers, if available, in all cases throughout the manuscript.

(3) Comparative genomic analysis: The strong point of the manuscript is the comparative genomic analysis between the phytophagous and predatory members of Pentatomoidae and not the phylogenetic analysis. This is because the paraphyly of subfamilies in the family Pentatomoidae is already well known and the results obtained in the present study are similar to earlier reports. So, in the title, rather than talking about the implications of the study with respect to phylogeny, authors should focus more on the comparative genomics of phytophagous and predatory members of Pentatomoidae. Accordingly, authors should revise the introduction by providing more information about the phytophagous and predatory members of Pentatomoidae. Further, authors should increase the taxonomic sampling for predatory members of Pentatomoidae based on mitogenome information available on NCBI GenBank database and redo the analysis. In the phylogenetic tree, authors can highlight the phytophagous and predatory members of Pentatomoidae separately.

(4) Manuscript preparation: In the results authors should first describe the mitogenome of Aelia fieberi, including information on protein coding genes, codon usage, tRNA, etc. (make sure that current fig 6 is after fig 1) and then provide phylogeny and comparative genomics between phytophagous and predatory members of Pentatomoidae in separate subheadings.

(5) Phylogenetic analysis: Provide maximum likelihood analysis along with Bayesian analysis.

(6) Focus on phylogeny: As stated earlier, authors should remove the focus on phylogenetic relationships among Pentatomoidae as it is not the strong point of the manuscript. Statements such as, "The mitogenomic information of A. fieberi could shed light on the evolution of Pentatomoidae" make little sense because the resulrts shown by authors are same as what earlier studies have shown and the additional sequence of Aelia fieberi mitogenome does not provide any new insights on the issue. What authors should focus on is the comparative genomices. The importance of the mitogenome of Aelia fieberi can be explained by suggesting filling the knowledge gap for this important crop pest.

Reviewers' comments:

Reviewer's Responses to Questions

**Comments to the Author**

1. Is the manuscript technically sound, and do the data support the conclusions?

Reviewer #1: Partly

Reviewer #2: Yes

2. Has the statistical analysis been performed appropriately and rigorously? 

Reviewer #1: Yes

Reviewer #2: Yes

3. Have the authors made all data underlying the findings in their manuscript fully available?

Reviewer #1: Yes

Reviewer #2: Yes

4. Is the manuscript presented in an intelligible fashion and written in standard English?

Reviewer #1: Yes

Reviewer #2: Yes

5. Review Comments to the Author

Reviewer #1: This paper reports the mitochondrial genome data of Aelia fieberi, which can provide some data support for the phylogeny of Pentatomidae or Pentatomoidea. However, the content of the paper is relatively simple, and the analysis method is not innovative, the analysis of the results is also innovative.

Reviewer #2: Review of manuscript Mitogenome of the stink bug Aelia fieberi (Hemiptera: Pentatomidae) and a phylogenetic analysis of Pentatomidae (PONE-D-23-11789)

In this study, authors sequenced and analyzed the complete mitogenome of Aelia fieberi. Similarly, authors compared the mitogenome of Aelia fieberi with 53 available complete mitogenomes of bugs of the Pentatomidae family. The analysis revealed that phytophagous bugs and predatory bugs had different evolutionary rates. Analysis also revealed that Asopinae and Phyllocephalinae are monophyly and Pentatominae and Podopinae are not monophyly groups. Authors suggested that the phylogenetic relationships of Pentatomoidae are complex, and need revaluation and revision. The current study represents the comparative mitogenomic and phylogenetic analyses within the Pentatomoidae family and provides better insights into the mitogenome features and evolution of bugs belonging to the Pentatomoidae family.

The overall study is original, relevant, and interesting but requires improvements in data analysis, results, and discussion sections of the manuscript. The statistical analysis and interpretations based on genetic analysis are appropriate. However, in-depth analysis, such as usage bias of start and stop codons, non-synonymous/synonymous ratios (ω = dN/dS ) in protein-coding genes of mitogenomes, gene rearrangement, and the divergence time analysis of Pentatomidae bugs will strengthen the results and discussion section of the manuscript. Generally, a strong codon usage bias is reported in Hemiptera. Authors should comment on an effective number of codon (ENC) values for Aelia fieberi and compare them with the other hemipteran species. Standard ENC curve (ENC vs G/C composition of the 3rd position of codons), will provide better information about the codon usage patterns in Hemipteran species.

6. PLOS authors have the option to publish the peer review history of their article (what does this mean?). If published, this will include your full peer review and any attached files.

Reviewer #1: No

Reviewer #2: **Yes: **Mandar S. Paingankar

---

## [Author Response · Author response to Decision Letter 0]

6 Sep 2023

We revised the manuscript according to the comments from reviewers.

---

## [Editor Report · Decision Letter 1]

28 Sep 2023

Mitogenome of the stink bug Aelia fieberi (Hemiptera: Pentatomidae) and a comparative genomic analysis between phytophagous and predatory members of Pentatomidae

PONE-D-23-11789R1

Dear Dr. Wang,

We’re pleased to inform you that your manuscript has been judged scientifically suitable for publication and will be formally accepted for publication once it meets all outstanding technical requirements.

Kind regards,

Neelesh Dahanukar, Ph.D.

Academic Editor

PLOS ONE

Additional Editor Comments:

Authors have made substantial changes to the manuscript as per the comments on the earlier draft. Authors have taken cognizance of all the comments and the revised manuscript reads well.

---

## [Editor Report · Acceptance letter]

2 Oct 2023

PONE-D-23-11789R1 

Mitogenome of the stink bug *Aelia fieberi* (Hemiptera: Pentatomidae) and a comparative genomic analysis between phytophagous and predatory members of Pentatomidae 

Dear Dr. Wang:

I'm pleased to inform you that your manuscript has been deemed suitable for publication in PLOS ONE. Congratulations! Your manuscript is now with our production department. 

Kind regards, 

on behalf of

Dr. Neelesh Dahanukar 

Academic Editor

PLOS ONE